# Epidemiology of Surgical Site Infections and Non-Surgical Infections in Neurosurgical Polish Patients—Substantial Changes in 2003–2017

**DOI:** 10.3390/ijerph16060911

**Published:** 2019-03-13

**Authors:** Małgorzata Kołpa, Marta Wałaszek, Anna Różańska, Zdzisław Wolak, Jadwiga Wójkowska-Mach

**Affiliations:** 1State Higher Vocational School in Tarnów, St. Luke’s Provincial Hospital in Tarnów, 33-100 Tarnów, Poland; malgorzatakolpa@interia.pl (M.K.); mz.walaszek@gmail.com (M.W.); zdzich_w@interia.pl (Z.W.); 2Department of Microbiology, Faculty of Medicine, Jagiellonian University Collegium Medicum, 31-121 Kraków, Poland; a.rozanska@uj.edu.pl

**Keywords:** neurosurgery, surgical site infections, spinal surgery, craniotomy, laminectomy

## Abstract

*Introduction*: The objective of the analysis was to determine the epidemiology of healthcare-associated infections (HAIs) in neurosurgical patients, paying special attention to two time points, 2003 and 2017, in order to evaluate the effectiveness of a surveillance program introduced in 2003 and efforts to reduce infection rates. *Materials and methods*: Continuous surveillance during 2003–2017 carried out using the HAI-Net methodology allowed us to detect 476 cases of HAIs among 10,332 patients staying in a 42-bed neurosurgery unit. The intervention in this before–after study (2003–2017) comprised standardized HAI surveillance with regular analysis and feedback. *Results*: The HAI incidence during the whole study was 4.6%. Surgical site infections (SSIs) accounted for 33% of all HAIs with an incidence rate of 1.5%. The remaining infections were pneumonia (1.1%) and bloodstream infections (0.9%). The highest SSI incidence concerned spinal fusion (FUSN, 2.2%), craniotomy (1.9%), and ventricular shunt (5.1%) while the associated total HAI incidence rates were 4.1%, 8.0%, and 18.6%, respectively. A significant reduction was found in HAI incidence between 2003 and 2017 in regard to the most common surgery types: laminectomy (4.5% vs. 0.8%); FUSN (11.8% vs. 0.8%); and craniotomy (10.1% vs. 0.4%). Significant changes were also achieved in selected elements of the unit’s work: pre-hospitalization duration, hospital stay, and surgery length reductions. Simultaneously, the general condition of patients became significantly worse: there was an increase in patients’ age and decreases in their general condition as expressed by ASA scores (The American Society of Anesthesiologists physical status classification system). *Conclusions*: HAI epidemiology changed substantially during the study period. Among the main types of HAI, SSIs were slightly predominant, but non-surgical HAIs accounted for almost two thirds of all infections; this indicates the need for surveillance of infection types other than SSIs in surgical patients. The implementation of active surveillance based on regular analysis and feedback led to a significant reduction in HAI incidence.

## 1. Introduction

Healthcare-associated infections (HAIs) and, particularly, the associated problem of antibiotic resistance are indicated by WHO as one of the most important challenges of modern medicine. WHO recommend a multimodal strategy for implementing infection prevention and control activities. This consists of several elements or components: monitoring practices, processes and outcomes, providing data feedback, and education of health care workers, among others. Monitoring or wider surveillance—the systematic collection, analysis, and interpretation of health data—is essential for the planning, implementation, and evaluation of public health practice. Health care facility surveillance should be based on standard definitions and methods and provide information for describing the epidemiology and microbiology of HAIs [1]. The scope of this control may vary, as well as the methods. The European Centre for Disease Prevention and Control (ECDC) recommends active surveillance of infections in intensive care units and of surgical site infections (SSIs) in selected surgical procedures [2]. Consequently, the epidemiological situation is well known and described regarding these recommended forms of surveillance. However, a thorough analysis of the data coming from an active but also comprehensive program of HAI surveillance in a Polish multi-profile hospital has demonstrated that active surveillance should additionally encompass patients from departments other than intensive care units and, depending on the needs of a given facility, those undergoing different surgical procedures [3]. This is particularly important in poorly described neurosurgery: studies on HAIs in neurosurgery are rarely undertaken or published, including in Poland.

Nevertheless, there are limited data available on both the incidence and burden of HAIs (all types of HAIs) in neurosurgical patients of hospitals in Central Europe, but several reports from different parts of the world confirm that the risk of HAIs in study populations is high, reaching 6.7% in India and 3.6% in a Turkish neurosurgical intensive care unit (ICU) [4,5]. Patients requiring neurosurgical treatment are at a higher risk of developing HAIs, both surgical site infections and other forms, due to the risk of impaired protective reflexes, increased application of invasive medical devices, longer hospital stay, and altered consciousness [4,6].

Carrying out continuous, targeted surveillance is an essential element that allows us to determine the current epidemiological situation of the unit and implement preventive actions. Therefore, the objective of this analysis was to determine both the epidemiology and microbiology of neurosurgical patients’ infections, as well as trends in their occurrence between 2003 (the beginning of HAI surveillance) and 2017, in order to evaluate the effectiveness of measures implemented in the field of infection prevention.

## 2. Materials and Methods

### 2.1. Setting

An intervention study using surveillance of HAIs, observation, and feedback was carried out in a neurosurgical unit in St. Luke Provincial Hospital in Tarnów, Poland, in 2003–2017. The department offers 42 hospital beds (including 6 intensive care beds with mechanical ventilation). Patients in very poor clinical condition do not stay in this department as they are generally sent to a separate general intensive care unit. Active surveillance of infections was implemented in the hospital in 2001, and the experiences concerning the neurosurgery unit were already the subject of previous general analyses not including trend analysis or detailed analyses of various HAI clinical forms [7]. The Infection Control Team (ICT) consists of a doctor and four full-time epidemiological nurses.

### 2.2. Outcome Measurement Methodology

The data analyzed the time when the studied unit introduced targeted and active HAI surveillance, initially (2003–2012), using tools, definitions, and protocols in accordance with the National Healthcare Safety Network (NHSN) [8]. Then, from 2012, the HAI recognition methodology and HAI record-keeping followed the Surveillance Network (HAI-Net) by the European Centre for Disease Prevention and Control (ECDC) [9,10]. For the purposes of this analysis, HAI cases originally qualified according to the NHSN criteria were retrospectively subjected to reclassification according to the ECDC definitions; hence, all HAI cases were qualified into individual HAI categories according to the ECDC case definition, keeping the divisions into catheter-related bloodstream infection (BSI) and BSI secondary to another infection, five subcategories of pneumonia (PN), and three forms of SSI. The surgeries performed were stratified by type of operation conforming to ICD 9-CM, according to the NHSN code (Table 1).

### 2.3. Bacterial Isolates

A single strain was derived from the first sample collected in the case of the first episode of HAI, and to confirm a diagnosis of BSI, at least two blood samples were taken. The strains were identified using BD Phoenix NID panels from the automated Phoenix 100 Becton Dickinson Diagnostic System (Becton Dickinson, Warsaw, Poland) according to the manufacturer’s instructions. All materials taken from gastrointestinal tract infections were tested for *Shigella* spp., *Salmonella* spp., rotavirus, norovirus, adenovirus, and *Clostridium difficile* (CD), and a general stool culture was performed.

### 2.4. Process Improvement Indicators

Beginning in 2003, changes were being implemented regarding HAI prevention and control by the ICT together with the staff of the departments (neurosurgery, operating room, and ICT); among other changes, these included the following:Hospital admission rules for shortening the pre-operative stay and optimal patient preparation for surgery to limit emergency surgery; preparation of the surgical team, includingdiagnostics and qualification for surgery as far as possible in an outpatient procedure without prior hospitalization before surgery,pre-operative screening at a preoperative assessment clinic and the decolonization of methicillin-resistant *Staphylococcus aureus* (MRSA) in elective procedures.Perioperative procedures for patient preparation for surgery, includinghair removal: cutting instead of shaving,bathing immediately prior to surgery,changing bed linens and patient’s clothing immediately before surgery.Work organization of the operating block, includingpreoperative checklist,surgical hand hygiene according to WHO guidelines,preparation of the operating field and surgical drape,application of antiseptic to the edges of the wound before sewing it.Patient care during the postoperative period:the five moments for hand hygiene,post-operative dressing and wound control.Active surveillance of all forms of HAIs:systematic collection, analysis, and interpretation of data for evaluation of practices,yearly feedback on the epidemiology and microbiology of HAIs,regular feedback on compliance with the procedures described above and hand hygiene.

In addition, regular education and training of health care workers based on the results of the surveillance were also implemented. Implementation of procedures and surveillance with analysis and feedback aimed at preventing SSIs were initiated by the ICT and concerned all operational procedures without distinguishing the type of surgery in accordance with global guidelines.

### 2.5. Statistical Analysis

In order to carry out an evaluation of the epidemiological situation, the following epidemiological indicator was applied: the cumulative incidence of HAI, defined as the number of HAI cases divided by the number of neurosurgery procedures × 100. Statistical analysis of the collected material employed IBM SPSS (SPSS—Statistical Package for the Social Sciences) software, STATISTICS 24, Armonk, NY, USA, and Microsoft Excel, Microsoft Office, 2016, Redmond, WA, USA. Statistical analysis was carried out with the use of basic statistical parameters, i.e., mean, standard deviation, 95% confidence intervals for the mean, and median. In order to compare the frequency of occurrence of qualitative feature variants, Pearson’s chi-square test of independence was applied; for variables in small quantities, Fisher’s exact test; and for quantitative variables, we used ANOVA. The level of significance was *p* < 0.05.

The use of data was approved by the Bioethical Committee of the Jagiellonian University (No. KBET/122.6120.118.2016 from May 25, 2016). All the data entered into the electronic database and analyzed in this study were previously anonymized. The work has been reported in line with the Standards for Quality Improvement Reporting Excellence (SQUIRE) criteria. The study was registered at http://www.clinicaltrials.gov, Identifier: NCT03686553.

## 3. Results

The total number of patients covered by the study amounted to 10,332 people operated upon in the neurosurgical unit in the study period of 15 years. In total, 476 HAIs were detected, and the total incidence rate was 4.6%. During the study (total, from 2003 to 2017) SSIs accounted for 33% of all HAIs: 157 cases (incidence 1.5%) (Table 2). The highest HAI incidence rates were observed with operations performed on the brain, i.e., ventricular shunt implantation surge (VP shunt, 77 cases, 18.6%) and craniotomy (235 cases, 8.0%). As for spinal surgery, the HAI incidence rates were significantly lower, i.e., 4.1% in spinal fusion (FUSN) (98 cases) and 1.4% in laminectomy (66 cases) (Table 2). Non-surgical infections accounted for 67% of all HAIs and were most often pneumonia (118 cases, 24.8% of all HAIs) and BSI (97, 20.4%). The PN incidence in patients needing mechanical ventilation (ventilator-associated pneumonia, VAP) was 40.5/1000 days of mechanical ventilation, and the incidence in patients with central line–associated bloodstream infections (CLA-BSI) was 3.3/1000 catheter days.

A significant reduction was found in HAI incidence before (2003) and after the implementation of surveillance (2017) regarding the most commonly performed types of procedures, i.e., laminectomy, 4.5% vs. 0.8% (relative risk (RR) 5.5, 95% confidence interval (CI) 1.57–19.16, *p* = 0.01); FUSN, 11.8% vs. 0.8% (RR 13.4, 95% CI 2.53–70.51, *p* < 0.001); craniotomy, 10.1% vs. 0.4% (25.3, 95% CI 2.87–170.58, *p* < 0.001); and VP shunt, 23.1% vs. 11.1% (1.8, 95% CI 0.43–8.24, *p* = 0.648) (Table 3).

As a result of the activities conducted, a significant change was achieved regarding selected elements of the unit’s work, i.e., there was a decrease in the duration of hospitalization before surgery (FUSN and craniotomy), as well as in the length of hospital stay (FUSN and craniotomy) and surgery duration (laminectomy and VP shunt), and the number of urgent surgeries was limited in favor of better preparation for planned procedures in craniotomy (Table 3). At the same time, there was a significant deterioration in the demographics of the patients operated: the patients’ age (laminectomy and craniotomy) and their general condition as expressed by ASA score (ASA 1–2 pts vs. ASA 3–4 pts) (laminectomy, FUSN, craniotomy) increased.

HAI epidemiology, unit work indicators, and patients’ characteristics following infrequently performed VP shunt procedures did not change (Table 3).

HAI microbiology was dominated by Gram-positive cocci (35.9%), mainly *Staphylococcus aureus* (120 cases, 25.2%), which was also the most common etiologic agent in SSIs (78 cases, 49.7%). In BSIs, coagulase-negative staphylococci (28 cases, 28.9%) were most frequently encountered. In pneumonia, *Acinetobacter baumannii* (26 isolates, 22.0%) was most often isolated. In total, 72 cases of HAIs (15.1%) lacked microbiological confirmation of various HAIs, and confirmation was obtained less frequently in the cases of pneumonia and gastrointestinal infection, of which only 70.3% and 27.3%, respectively, were microbiologically confirmed (Table 4).

## 4. Discussion

In the studied neurosurgical unit, SSI incidence amounted to 1.5% and was close to expectations; other authors report it at levels of 1.3% (95% CI 1.1–1.5) in Europe 2008–2009, with an inter-country range of 0.4%–6.3%, or 2.2% (RR 11.3, 95% CI 4.2–30.6; *p* < 0.01) in Italy in 2002–2004, among others [11,12,13,14]. Nevertheless, to confirm the effectiveness of the ICT and the sensitivity of infection prevention, it is relevant that SSI characteristics, i.e., the dominance of deep incisional infections, which accounted for more than half of the cases, are confirmed in the literature since a similar result of 62% of deep incisional infections among all SSIs was obtained by Smith [15].

In the period studied, the majority of surgeries were performed within the spine, i.e., laminectomy or FUSN. The SSI incidence in laminectomy amounted to 0.6% and was one of the lowest among the European countries that implement the European HAI-Net program (mean 1.3%, ranging from 0.4% in Portugal to 6.3% in Hungary) [16]. Notably, during the implementation of surveillance for this procedure type (between 2003–2017), HAI risk, including SSI, was significantly reduced. On the other hand, analysis of patient demographics reveals some differences between the study population and patients in Europe: the age of operated patients was lower than the age of operated patients in HAI-Net (49 years vs. 54 years). This is probably due to using less strict criteria for surgical interventions or to the health condition of Polish patients who require surgical intervention earlier than is the case in other European countries [16], which appears to be associated with better functional capacities of younger patients, reducing susceptibility to SSI, resulting in low incidence.

Research into SSIs following spinal fusion (FUSN) indicates a high similarity to the data presented by other authors. In American multicenter NHSN studies from 2006 to 2008, values close to ours were obtained, i.e., the median ranged from 0.7% to 4.2% [17,18], despite the fact that single-center data quote incidence rates reaching 12% [15,19,20]. It was no different in the case of craniotomy, for which in an NHSN program (data from 2006–2008), the median incidence was 2.9% [17], although Zhan et al. gave a much higher value of 7.4% [21].

However, following the rarely performed VP shunt surgeries, SSI incidence in our study was high and corresponded to the incidence in patients with multiple risk factors described in the NHSN program, in which it reached 5.9% [17]; nevertheless, an even higher incidence is reported by Rosenthal, who, on the basis of an analysis of data from 30 countries around the world, including from America, Asia, Africa, and Europe, reported values as high as 12.9% [22].

The implementation of surveillance and infection prevention and control was of paramount importance as it resulted in both measurable improvements in the organization of the unit and functioning of the operating block and a reduction in the risk of HAI infections of all clinical forms, in all procedures—with the exception of rarely performed VP shunts. This was particularly expressed in the significant reduction in the durations of operations, e.g., in laminectomy; however, in 2017, this value was much greater than the 75 min given in the European HAI-Net program [16], which may be suggestive of a necessity to conduct further intensive works in this respect. Also, the length of hospitalization obtained after the surveillance period (9 or 11 days, depending on the procedure type) is still unsatisfactory—it is longer than, e.g., in Holland, where it is 6 days on average [23].

Surgery duration is pointed out as one of the most important parameters and a significant risk factor for neurosurgical operations [24,25]. Therefore, given the importance of SSIs concerning patient outcomes and health care economics, surgeons should focus their efforts on reducing operative time [26].

It is all the more significant that in the patient population observed, the characteristics of patients changed significantly, especially regarding their age, which may indicate a trend of an aging patient population and their increased susceptibility to infections. At the same time, the next property that a researcher has no control over is sex: among the patients operated on, men were predominant, and they are also usually more vulnerable to infections after neurosurgical procedures [27,28]. The fact that being a man greatly affects the risk of SSI was also reported in Poland for thoracic procedures [29].

Owing to a comprehensive analysis of HAI data in the study population, it became clear that non-surgical infections are very important and equally common among neurosurgical patients undergoing surgery; among them, there were PN and BSI, which accounted for almost 2/3 of all HAIs, especially following craniotomy. Particularly, VAP incidence was higher than expected: in neurosurgical intensive care units in Italy, it was 4-fold lower [13], and in the NNIS program in 2006–2008, it was 8-fold lower [17]. The situation is slightly better for CLA-BSI incidence, which is identical to that observed in American neurosurgical ICUs [18]; however, it is simultaneously about two times higher than in Italy [13]. Such a comparison between the VAP incidence in the studied population and those in other neurosurgery ICUs could be affected by the type and severity of treated neurosurgical diseases. Thus, drawing of conclusions should be done with caution due to this limitation.

In our study, it is important to pay attention to the fact that in as many as 40% of BSI cases, it was impossible to confirm the origin of infection. This situation may suggest a lack of an appropriate approach to BSI diagnostics, which is unfortunately in line with the Polish national phenomenon, because, even though we present single-center studies, a similar situation was confirmed in relation to patients of Polish ICUs in a multicenter study [30]. The remaining types of infections were observed relatively less frequently; however, a large number of non-microbiologically-confirmed gastroenteritis cases, over 2/3, draws our attention.

In an analysis of etiologic agents of infection, according to our expectations, it was confirmed that BSIs were most often caused by *Staphylococcus aureus* and coagulase-negative staphylococci. Pneumonia was most frequently brought about by *Acinetobacter baumannii*, but it should be noted that as many as 54 (31%) cases of PN were not confirmed microbiologically. The situation is similar to the one described in a multicenter study of Polish ICUs which showed the weaknesses of microbiological diagnostics or lack of access to it [31]. *Acinetobacter baumannii* is not a common cause of pneumonia in most European countries [32], but in recent years in Polish ICUs, a large burden of pneumonia by Gram-negative bacilli has been documented with dominance of *A. baumannii*, which was responsible for as many as 1 in 5 cases [31]. *A. baumannii* is not commonly regarded as a major pneumonia pathogen, and because only a minority of pneumonia microbiological diagnoses were properly made in the studied unit and also in many Polish ICUs, it is difficult to know whether the high prevalence of *A. baumannii* may have resulted from inadequate diagnosis or environmental contamination, e.g., stethoscopes can be a cause of cross-infection [33]. On the other hand, other authors have also reported a high proportion of non-fermenting bacilli in infections in neurosurgical ICUs [34] and in other surgical departments [35].

In SSIs, *Staphylococcus aureus* was predominant, and *Escherichia coli* was for urinary tract infections. *Clostridium difficile* was seemingly dominant in gastrointestinal tract infections; however, in as many as 63% of these infections, no infection factor was detected, or no such examinations were carried out despite the availability of a full package of microbiological and virologic tests for diagnosing these infections.

There are some limitations of the present study. Firstly, the research involves only one center. Secondly, in the period studied, the infection detection method and compliance with the prevention procedure were not systematically validated. Additionally, in 2012, the NHSN protocol changed to ECDC; however, the authors’ assessment is that this did not influence the number of bloodstream infections, pneumonia cases, and surgical site infections detected or qualified. It could only influence the number of urinary tract infections qualified. The other limitation is the organization of the unit’s work and the rule that patients in very poor clinical condition do not stay in this department as they are generally sent to an intensive care unit; those patients were excluded from the analysis.

However, on the other hand, this study is unique regarding the reporting of HAIs in neurosurgery with respect to Polish patients. The results of the analysis allow us to define the strengths and weaknesses of infection surveillance and its priorities for the future. Firstly, intervention is required regarding the high incidence of non-surgical infections, especially pneumonia and VAP, as well as microbiological diagnostics of VAP. Secondly, it is essential to undertake actions to improve the diagnostics of hospital-acquired bloodstream infections, pneumonia, and gastrointestinal tract infections. The results presented show considerable (significant) differences in terms of both epidemiological indicators and patient characteristics and the organization of the unit’s work between 2003 and 2017, which indicate that the surveillance of infections requires continuity and obtaining our own reliable epidemiological data.

## 5. Conclusions

To conclude, the study has validated the usefulness of active surveillance of HAIs in neurosurgical patients based on regular analysis and feedback. We suggest the implementation of infection control and prevention based on evidence-based medicine. Such active SSI surveillance should be recommended by the ECDC for the everyday activity of ICTs and should be adopted by other hospitals in Poland.

## Figures and Tables

**Table 1 ijerph-16-00911-t001:** List of surgeries and ICD-9 codes.

Code	Operative Procedure	ICD-9
LAM	Laminectomy: exploration or decompression of spinal cord through excision or incision into vertebral structures	03.01; 03.02; 03.09; 80.50; 80.5; 80.53; 80.54; 80.59; 84.60–84.69; 84.80–84.85.
FUSN	Spinal implant surgery: spinal fusion, Immobilization of spinal column	81.00–81.08
CRAN	Craniotomy: incision through the skull to excise, repair, or explore the brain; does not include taps or punctures	01.12; 01.14; 01.20–01.25; 01.28; 01.29; 01.31; 01.32; 01.39; 01.41; 01.42; 01.51–01.53; 01.59; 02.11–02.14; 02.91–02.93; 07.51–07.54; 07.59; 07.61–07.65; 07.68; 07.69; 07.71; 07.72; 07.79; 38.01; 38.11; 38.31; 38.41; 38.51; 38.61; 38,81; 39.28.
VP Shunt	VP shunt: ventricular shunt operations, including revision and removal of shunt	02.21; 02.22; 02.31–02.35; 02.39; 02.42; 02.43; 54.95.

**Table 2 ijerph-16-00911-t002:** Incidence of hospital-acquired infections (HAIs) by major and specific site of infection in the years 2003–2017.

Surgery Type (*N* = 10,332)		LAM (*n* = 4571)	FUSN (*n* = 2397)	CRAN (*n* = 2950)	VP Shunt (*n* = 414)	Incidence	Share in the Total Pool of HAIs (%)
All HAIs (*n* = 476)		66	98	235	77		
HAI Incidence (%)		1.4	4.1	8.0	18.6	4.6	
Surgical site infection (*n* = 157)		SSI incidence (%)					33.0
		0.6	2.2	1.9	5.1	1.52	
superficial incisional (*n* = 56)		11	21	22	2	0.54	
deep incisional (*n* = 90)		16	31	30	13	0.87	
organ/space (*n* = 11)		0	0	5	6	0.11	
Pneumonia (PN) (*n* = 118)		PN incidence (%)					24.8
		0.1	0.4	2.7	5.6	1.14	
PN1 (*n* = 2)		0	0	2	0	0.02	
PN2 (*n* = 8)		2	0	1	5	0.08	
PN3 (*n* = 0)		0	0	0	0	0.00	
PN4 (*n* = 70)		4	6	46	14	0.68	
PN5 (*n* = 38)		0	3	31	4	0.37	
Bloodstream infections (BSI) (*n* = 97)		BSI incidence (%)					20.4
		0.2	0.8	1.8	4.1	0.94	
BSI: catheter related (*n* = 23)		0	1	17	5	0.22	
BSI: unknown origin (*n* = 44)		5	14	18	7	0.43	
sepsis (*n* = 3)		0	0	3	0	0.03	
BSI: secondary (*n* = 27)		4	3	15	5	0.26	
Urinary tract infection (*n* = 63)		UTI incidence (%)					13.2
		0.3	0.3	0.9	3.1	0.61	
microbiologically confirmed	yes (*n* = 54)	8	6	27	13	0.52	
	no (*n* = 9)	6	2	1	0	0.09	
Gastrointestinal (*n* = 33)		GI incidence (%)					6.9
		0.2	0.4	0.4	0.7	0.32	
*Clostridium difficile* infection (*n* = 9)		2	1	4	2	0.09	
Gastroenteritis (*n* = 24)		6	8	9	1	0.23	
Skin and soft tissue infection (*n* = 8)		SST incidence (%)					1.7
		0.0	0.0	0.1	0.0	0.08	
skin infection (*n* = 8)		2	2	4	0	0.08	

Legend: LAM, laminectomy; FUSN, spinal implant surgery; CRAN, craniotomy; VP shunt, ventricular shunt implantation surgery; PN1, positive quantitative culture of material from minimally contaminated of secretions from the lower respiratory tract (LRT); PN2, positive quantitative culture of material from possibly contaminated LRT; PN3, pneumonia confirmed microbiologically using alternative methods; PN4, positive qualitative culture of sputum or secretions from the respiratory tract; PN5, non-microbiologically-confirmed pneumonia.

**Table 3 ijerph-16-00911-t003:** Characteristics of patients (with and without HAIs) and their hospitalization in two phases of the observational period: before (2003) and after (2017) implementation of active surveillance of hospital-acquired infections (HAIs).

Surgery Type	Spinal Surgery				Brain Surgery			
	LAM		FUSN		CRAN		VP Shunt	
Phase of the Study	Before	After	Before	After	Before	After	Before	After
surgeries, no.	133	504	34	252	99	240	13	27
HAI, no.	6	4	4	2	10	1	3	3
HAI incidence (%)	4.5	0.8	11.8	0.8	10.1	0.4	23.1	11.1
RR 95%CI, Fisher’s exact test (*p*)	5.5, 1.57–19.16, *p* = 0.01		13.4, 2.53–70.51, *p* < 0.001		25.3, 2.87–170.58, *p* < 0.001		2.1, 0.43–8.24, *p* = 0.648	
Patient age (years)								
Mean (SD)	48 (13.8)	52 (14.5)	52 (13.1)	51 (14.5)	56 (16.9)	61 (16.4)	57 (14.7)	54 (20.4)
ANOVA (*p*)	*p* = 0.005		*p* = 0.647		***p* = 0.031**		*p* = 0.611	
Hospitalization duration prior to surgery (days)								
Mean (SD)	6 (5.041)	3 (2.696)	8 (6.569)	4 (4.022)	4 (5.9)	3 (4.8)	9 (9.8)	9 (12.6)
ANOVA (*p*)	*p* < 0.001		*p* < 0.001		*p* = 0.150		*p* = 0.857	
Hospitalization duration (days)								
Mean (SD)	13 (8.3)	9 (5.4)	20 (21.5)	11 (12.1)	14 (13.8)	13 (10.8)	19 (14.9)	20 (22.2)
ANOVA (*p*)	*p* < 0.001		*p* < 0.001		*p* = 0.406		*p* = 0.885	
Surgery duration (minutes)								
\Mean (SD)	127 (55.5)	116 (55.6)	135 (39.6)	141 (65.7)	110 (46.3)	88 (56.4)	80 (21.7)	65 (31.1)
ANOVA (*p*)	*p* = 0.043		*p* = 0.027		*p* = 0.001		*p* = 0.043	
Sex								
Men	85 (63.9%)	280 (55.6%)	17 (50.0%)	138 (54.8%)	66 (66.7%)	147 (61.2%)	8 (61.5%)	16 (59.3%)
Women	48 (36.1%)	224 (44.4%)	17 (50.0%)	114 (45.2%)	33 (33.3%)	93 (38.8%)	5 (38.5%)	11 (40.7%)
Fisher’s exact test (*p*)	*p* = 0.051		*p* = 0.366		*p* = 0.208		*p* = 0.585	
Operation mode								
Planned	132 (99.2%)	471 (93.5%)	34 (100.0%)	234 (92.9%)	23 (23.2%)	107 (44.6%)	13 (100.0%)	21 (77.8%)
Urgent	1 (0.8%)	33 (6.5%)	0 (0.0%)	18 (7.1%)	76 (76.8%)	133 (55.4%)	0 (0.0%)	6 (22.2%)
Fisher’s exact test (*p*)	*p* = 0.003		*p* = 0.095		*p* < 0.001		*p* = 0.077	
Patient condition according to the ASA score								
ASA 1–2 pts	61 (45.9%)	159 (31.5%)	16 (47.1%)	88 (34.9%)	20 (20.2%)	23 (9.7%)	1 (7.7%)	1 (3.7%)
ASA 3–5 pts	72 (54.1%)	345 (68.5%)	18 (52.9%)	164 (65.1%)	79 (79.8%)	215 (90.3%)	12 (92.3%)	26 (96.3%)
Fisher’s exact test (*p*)	*p* < 0.001		*p* = 0.008		*p* < 0.001		*p* = 0.978	

**Table 4 ijerph-16-00911-t004:** Etiology of different types of HAIs, 2003–2017.

Microorganism	BSI	GI	PN	SSI	SST	UTI	Total
	*n* (%)	*n* (%)	*n* (%)	*n* (%)	*n* (%)	*n* (%)	*n* (%)
**Gram-positive cocci**							
*Staphylococcus aureus*	24 (24.7)	0 (0.0%)	13 (11.0)	78 (49.7)	4 (50.0)	1 (1.6)	120 (25.2)
*Coagulase-negative staphylococci*	28 (28.9)	0 (0.0%)	0 (0.0%)	5 (3.2)	1 (12.5)		34 (7.1)
*Streptococcus* spp.	0 (0.0%)	0 (0.0%)	5 (4.2)	3 (1.9)	0 (0.0%)	9 (14.3)	17 (3.6)
**Enterobacteriaceae**							
*Escherichia coli*	10 (10.3)	0 (0.0%)	5 (4.2)	9 (5.7)	0 (0.0%)	19 (30.2)	43 (9.0)
*Klebsiella* spp.	7 (7.2)	0 (0.0%)	13 (11.0)	2 (1.3)	0 (0.0%)	7 (11.1)	29 (6.1)
*Enterobacter* spp.	7 (7.2)	0 (0.0%)	5 (4.2)	18 (11.5)	1 (12.5)	1 (1.6)	32 (6.7)
*Proteus* spp.	2 (2.1)	0 (0.0%)	7 (5.9)	0 (0.0%)	0 (0.0%)	5 (7.9)	14 (2.9)
*Serratia* spp.	3 (3.1)	0 (0.0%)	0 (0.0%)	2 (1.3)	0 (0.0%)		5 (1.1)
**Non-fermenting Gram-negative bacteria**							
*Acinetobacter baumannii*	11 (11.3)	0 (0.0%)	26 (22.0)	24 (15.3)	1 (12.5)	4 (6.3)	66 (13.9)
*Pseudomonas aeruginosa*	0 (0.0%)	0 (0.0%)	6 (5.1)	7 (4.5)	1 (12.5)	6 (9.5)	20 (4.2)
*Morganella morganii*	0 (0.0%)	0 (0.0%)	3 (2.5)	1 (0.6)	0 (0.0%)	1 (1.6)	5 (1.1)
**Others**							
*Clostridium difficile*	0 (0.0%)	9 (27.3)	0 (0.0%)	0 (0.0%)	0 (0.0%)	0 (0.0%)	9 (1.9)
*Candida* spp.	2 (2.1)	0 (0.0%)	0 (0.0%)	0 (0.0%)	0 (0.0%)	8 (12.7)	10 (2.1)
Non-microbiologically-confirmed	3 (3.1)	24 (72.7)	35 (29.7)	8 (5.1)	0 (0.0%)	2 (3.2)	72 (15.1)
Total	97 (100)	33 (100)	118 (100)	157 (100)	8 (100)	63 (100)	476 (100)

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
