# Peer review of "Epidemiology of Surgical Site Infections and Non-Surgical Infections in Neurosurgical Polish Patients—Substantial Changes in 2003–2017"

_ijerph, 2019, doi:10.3390/ijerph16060911_

Round 1

Reviewer 1 Report

The Authors present the results of an interesting analysis of the epidemiology of
hospital-acquired infections in neurosurgical patients from 2003 to 2017.

Major comments:
The Authors are kindly invited to perform extensive stylistic and grammar editing of the
paper.

Minor comments:
-In the abstract, the authors should mention that practice changes were implemented in
2003, on order to reduce the incidence of HAIs

- In the abstract the research question is not clearly stated in the introduction

- The abstract reports some abbreviations without explanation

- Line 28: “the demographics of the patients deteriorated significantly”. This statement
is unclear

- Lines 67-68 “Patients in very poor clinical condition do not stay in this department as
they are generally sent to a separate general intensive care unit “. If those patients were
excluded by the analysis, this could be a selection bias. Please acknowledge that in the
limitations of the study

Lines 73-74” Beginning in 2003, changes were being implemented as regards HAI
prevention and control by the ICT together with the staff of the departments
(neurosurgery, operating room and ICT), which encompassed, among others:… “ The
Authors should describe more precisely the measures that were taken in neurosurgical
patients. Specifically, the Authors identify 4 procedure categories (craniotomy,
ventriculoperitoneal shunt, laminectomy, spinal fusion). Please, explain if specific
infection prevention measures for each of these procedure categories were applied. This
could be an important learning point for the reader and should be also reported both in
the methods and in the discussion.

-In the methods and table 1, the Authors group the neurosurgical procedures in 4
different categories (LAM, FUSN, CRAN, VSHN). There are two main concerns. The first
is that these categories include only some, but not all, the procedures that are performed
in a modern neurosurgical department. As an example, endovascular procedures for
brain aneurysms and AVMs are not considered, at all, although those patients may need
ICU admission. The second concern is that the four categories could carry different risk
of infection depending on the type and severity of treated pathology and on patients’
neurological status. Thus, over the years, the type of neurosurgical diseases and of
surgeries performed within a single category could have changed and this could have
affected the risk of morbidity and of infections within that specific category.
The type of neurosurgical diseases and potential changes in the epidemiology of the
treated diseases should be considered in the manuscript, because those can affect the
validity of the results

-Discussion, lines 206-209: “Particularly, VAP incidence was higher than expected, e.g. in
neurosurgical intensive care units in Italy, it was 4-fold lower [11], and in the NNIS
programme in 2006–2008, it was even 8-fold lower [15]. It is slightly better as regards
CLA-BSI incidence, which is identical to that observed in the American neurosurgical
ICUs [16], however, it is simultaneously about 2x higher than in Italy [11]. “ The
comparison between the incidence of VAP incidence in the analyzed ICU unit with
respect to other ICUs in the world could be affected by the type and severity of treated
neurosurgical diseases. Thus, the comparison could be not feasible. Moreover, the
authors report that: “Patients in very poor clinical condition do not stay in this
department as they are generally sent to a separate general intensive care unit “. Please
clarify.

Author Response

DETAILED RESPONSE TO REVIEWERS,

STEP-BY-STEP REPLIES TO REVIEWERS' COMMENTS:

 Reviewer #1:

The Authors present the results of an interesting analysis of the epidemiology of hospital-acquired infections in neurosurgical patients from 2003 to 2017.

Major comments: The Authors are kindly invited to perform extensive stylistic and grammar editing of the paper.

Authors’ reply: Corrected according to suggestions – the manuscript was corrected by English editing service.

Minor comments:

-In the abstract, the authors should mention that practice changes were implemented in 2003, on order to reduce the incidence of HAIs

- In the abstract the research question is not clearly stated in the introduction

Authors’ reply: Corrected according to suggestions (lines 13-16), as below:

(...) The objective of the analysis was to determine the epidemiology of healthcare-associated infections (HAIs) in neurosurgical patients, paying special attention to two time points, 2003 and 2017, in order to evaluate the effectiveness of a surveillance program introduced in 2003 and efforts to reduce infection rates. (...)

Authors’ reply: Corrected according to suggestions.

- The abstract reports some abbreviations without explanation;

Authors’ reply: Corrected according to suggestions.

- Line 28: “the demographics of the patients deteriorated significantly”. This statement is unclear

Authors’ reply: Corrected according to suggestions (lines 28-31), as below:

“(...) Simultaneously, the general condition of patients became significantly worse: there was an increase in patients’ age and decreases in their general condition as expressed by ASA scores (The American Society of Anesthesiologists physical status classification system). (...)”

- Lines 67-68 “Patients in very poor clinical condition do not stay in this department as they are generally sent to a separate general intensive care unit “. If those patients were excluded by the analysis, this could be a selection bias. Please acknowledge that in the limitations of the study

Authors’ reply: Corrected according to suggestions (lines 282-284), as below:

(...) The other limitation is the organization of the unit's work and the rule that patients in very poor clinical condition do not stay in this department as they are generally sent to an intensive care unit; those patients were excluded from the analysis.. (...)

- Lines 73-74” Beginning in 2003, changes were being implemented as regards HAI prevention and control by the ICT together with the staff of the departments (neurosurgery, operating room and ICT), which encompassed, among others:… “ The Authors should describe more precisely the measures that were taken in neurosurgical patients. Specifically, the Authors identify 4 procedure categories (craniotomy, ventriculoperitoneal shunt, laminectomy, spinal fusion). Please, explain if specific infection prevention measures for each of these procedure categories were applied. This could be an important learning point for the reader and should be also reported both in the methods and in the discussion.

Authors’ reply: Corrected according to suggestions, (lines 104-135), as below:

(...) Beginning in 2003, changes were being implemented as regards HAI prevention and control by the ICT together with the staff of the departments (neurosurgery, operating room and ICT), which encompassed, among others:

Hospital admission rules for shortening the pre-operative stay and      optimal patient preparation for surgery to limit emergency surgery;      preparation of the surgical team, including

diagnostics and       qualification for surgery as far as possible in an outpatient procedure       without prior hospitalization before surgery,

pre-operative       screening at a preoperative assessment clinic and the decolonisation of       Methicillin-Resistant Staphylococcus aureus MRSA in elective procedures.

Perioperative      procedures for patient preparation for surgery, including

hair removal:       cutting instead of shaving,

bathing       immediately prior to surgery,

changing bed       linens and patient’s clothing immediately before surgery.

Work organization      of the operating block, including

preoperative       checklist,

surgical hand       hygiene according to WHO guidelines,

preparation of       the operating field and surgical drape,

application of       antiseptic to the edges of the wound before sewing it.

Patient care      during the postoperative period:

the 5 moments for       hand hygiene,

post-operative       dressing and wound control.

Active      surveillance of all forms of HAIs:

systematic       collection, analysis, and interpretation of data for evaluation of       practices,

yearly feedback       on the epidemiology and microbiology of HAIs,

regular feedback       on compliance with the procedures described above and hand hygiene.

In addition, regular education and training of health care workers based on the results of the surveillance were also implemented. Implementation of procedures and surveillance with analysis and feedback aimed at preventing SSIs were initiated by the ICT and concerned all operational procedures without distinguishing the type of surgery in accordance with global guidelines. (...)

These procedures may seems basic but in the local situation in 2003 it was a big challenge for the ward staff and ICT. As a results, In the analyzed period no procedures specific to particular types of procedures were introduced by ICT.

- In the methods and table 1, the Authors group the neurosurgical procedures in 4 different categories (LAM, FUSN, CRAN, VSHN). There are two main concerns. The first is that these categories include only some, but not all, the procedures that are performed in a modern neurosurgical department. As an example, endovascular procedures for brain aneurysms and AVMs are not considered, at all, although those patients may need ICU admission. The second concern is that the four categories could carry different risk of infection depending on the type and severity of treated pathology and on patients’ neurological status. Thus, over the years, the type of neurosurgical diseases and of surgeries performed within a single category could have changed and this could have affected the risk of morbidity and of infections within that specific category. The type of neurosurgical diseases and potential changes in the epidemiology of the treated diseases should be considered in the manuscript, because those can affect the validity of the results

Authors’ reply: The methodology of the presented analysis, including data gathering, is based on NHSN and HAI-Net ECDC protocols according to which patients are qualified into specific groups (categories) based on type of treatment, not the illness or general condition (however ASA scale serves – only – as a basic indicator of patient general condition). This approach gives the infection control team the opportunity to inspect the work of the department based on the analysis of the most specific problems and risk factors. As a result, the obtained data are unified, gathered according to objective, non-modifiable data collection and analysis systems, including patients' qualifications – into the exposed patients group (denominator) or patients with HAIs symptoms. Such stratification of patients enables conducting epidemiological analysis for inference (from detail to general) and determining priorities in infection control at the local level (hospital) as well as in relation to comparative data, benchmarking. In the 2016 „Guidelines on Core Components of Infection Prevention and Control Programmes at the National and Acute Health Care Facility Level (http://www.who.int/gpsc/core-components.pdf) WHO recommends Hospital-based infection surveillance systems linked to integrated public health infection surveillance systems (Core component 4. Surveillance). Such a unified approach also contributes to generalization, e.g. neurosurgical patients are generally patients who require surgery either in the brain or in the spinal cord area. This is a great simplification, but it enables ICT to prepare a set of regulations and procedures covering the scope of these two fundamentally different problems. On the other hand, in the studied unit of neurosurgery within 15 years of surveillance, 45 endovascular procedures for brain aneurysms and AVMs were performed, which constituted 1.5% of all cerebral surgery procedures, therefore an analysis concerning this selected disease entity at the hospital level is impossible – it includes small data set.

– Discussion, lines 206-209: “Particularly, VAP incidence was higher than expected, e.g. in neurosurgical intensive care units in Italy, it was 4-fold lower [11], and in the NNIS  in 2006–2008, it was even 8-fold lower [15]. It is slightly better as regards CLA-BSI incidence, which is identical to that observed in the American neurosurgical ICUs [16], however, it is simultaneously about 2x higher than in Italy [11].“ The comparison between the incidence of VAP incidence in the analyzed ICU unit with respect to other ICUs in the world could be affected by the type and severity of treated neurosurgical diseases. Thus, the comparison could be not feasible. Moreover, the authors report that: “Patients in very poor clinical condition do not stay in this department as they are generally sent to a separate general intensive care unit “. Please clarify.

Authors’ reply: Corrected according to suggestions, (lines 246-249), as below:

(…) Such a comparison between the VAP incidence in the studied population and those in other neurosurgery ICUs could be affected by the type and severity of treated neurosurgical diseases. Thus, drawing of conclusions should be done with caution due to this limitation. (…)

Reviewer 2 Report

Abstract:

Line 15-16: I would advise you to write “with special attention from 2003 to 2017”.

Line 18 “10,332”: it is not clear if all these patients have HAI or only a part of these. I would advise you to review the sentence.

Line 38 “drug resistence” it is maybe too generic, I would probably suggest “antibiotic resistance”

Introduction:

Line 41: I would insert a reference.

Line 43: I would insert a reference.

Methods:

It is not specified what type of study is involved.

Line 64-82: I would recommend doing a single paragraph with the name "Setting”

Results:

Line 122-124: this sentence is in contrast with line 18. Please revise it.

Line 132: “before”: the study takes places in the time frame 2003-2017. 2003 is not included and the comparison with it should be more appropriate comparing estimates and CI to assess differences or not

Table 3: the amount of women and men is 100,1%

Provide results estimates with confidence intervals.

Discussion:

Line 154-155: Please add IC to assess whether there are or not significant differences in comparisons.

Considering that the stethoscopes can be cause of cross-infection, I would suggest reading this article for a brief in-depth analysis:

“Tanning the bugs – a pilot study of an innovative approach to stethoscope disinfection” Messina G., Rosadini D., Burgassi S., Messina D., Nante N., Tani M., Cevenni G.  Journal of Hospital Infection, 2017, 95,2:228-30  doi:10.1016/j.jhin.2016.12.005

References:

Line 295: it is missed  ".eu"

Line 298: the link is not correct, you should write again with “Publications” and “HAI-ICU”

Line 317: the link is not correct, you should write again with “NHSN/PDFS/DATASTAT/NNIS_2004.PDF

Author Response

DETAILED RESPONSE TO REVIEWERS,

STEP-BY-STEP REPLIES TO REVIEWERS' COMMENTS:

Reviewer #2:

Line 15-16: I would advise you to write “with special attention from 2003 to 2017”.

Authors’ reply: Corrected according to suggestions, (lines 13-16), as below:

(...) The objective of the analysis was to determine the epidemiology of healthcare-associated infections (HAIs) in neurosurgical patients, paying special attention to two time points, 2003 and 2017, in order to evaluate the effectiveness of a surveillance program introduced in 2003 and efforts to reduce infection rates. (...)

Line 18 “10,332”: it is not clear if all these patients have HAI or only a part of these. I would advise you to review the sentence.

Authors’ reply: Corrected according to suggestions, (lines 17-18), as below:

(…) allowed us to detect 476 cases of HAIs among 10,332 patients (…)

Line 38 “drug resistance” it is maybe too generic, I would probably suggest “antibiotic resistance”

Authors’ reply: Corrected according to suggestions.

Introduction:

Line 41: I would insert a reference.

Authors’ reply: Corrected according to suggestions.

Line 43: I would insert a reference.

Authors’ reply: Corrected according to suggestions.

Methods:

It is not specified what type of study is involved.

Authors’ reply: Corrected according to suggestions, (lines 74), as below:

(…) An intervention study using surveillance of HAIs, observation, and feedback was carried out in a neurosurgery unit in St. Luke Provincial Hospital in Tarnów, Poland, in 2003–2017. (…)

Line 64-82: I would recommend doing a single paragraph with the name "Setting”

Authors’ reply: Corrected according to suggestions, as below:

Results:

Line 122-124: this sentence is in contrast with line 18. Please revise it.

Authors’ reply: Corrected according to suggestions

Line 132: “before”: the study takes places in the time frame 2003-2017. 2003 is not included and the comparison with it should be more appropriate comparing estimates and CI to assess differences or not

Authors’ reply: In the first paragraph of the Results section we present general epidemiological rates for the whole period, that is 15 years. In the second paragraph of the Results section we present incidence rates for 2003 (before implementation of preventive recommendations) and for 2017 (after), together with RR, CI and p-values. The same approach is used in Tables.

Table 3: the amount of women and men is 100,1%

Authors’ reply: Corrected according to suggestions.

Provide results estimates with confidence intervals.

Authors’ reply: Corrected according to suggestions.

Discussion:

Line 154-155: Please add IC to assess whether there are or not significant differences in comparisons.

Authors’ reply: Corrected according to suggestions, (lines 188-191), as below:

(…) In the studied neurosurgical unit, SSI incidence amounted to 1.5% and was close to expectations; other authors report it at levels of 1.3% (95%CI 1.1–1.5) in Europe 2008–2009, with an inter-country range of 0.4%–6.3%, or 2.2% (RR 11.3 95%CI 4.2–30.6; p < 0.01) in Italy in 2002–2004, among others [11,12,13,14]. (…)

Considering that the stethoscopes can be cause of cross-infection, I would suggest reading this article for a brief in-depth analysis: “Tanning the bugs – a pilot study of an innovative approach to stethoscope disinfection” Messina G., Rosadini D., Burgassi S., Messina D., Nante N., Tani M., Cevenni G.  Journal of Hospital Infection, 2017, 95,2:228-30  doi:10.1016/j.jhin.2016.12.005

Authors’ reply: Corrected according to suggestions, (lines 262-269), as below:

(…) Acinetobacter baumannii is not a common cause of pneumonia in most European countries [32], but in recent years in Polish ICUs, a large burden of pneumonia by Gram-negative bacilli has been documented with dominance of A. baumannii, which was responsible for as many as 1 in 5 cases [31]. A. baumannii is not commonly regarded as a major pneumonia pathogen, and because only a minority of pneumonia microbiological diagnoses were properly made in the studied unit and also in many Polish ICUs, it is difficult to know whether the high prevalence of A. baumannii may have resulted from inadequate diagnosis or environmental contamination, e.g., stethoscopes can be a cause of cross-infection [33]. (…)

References:

Line 295: it is missed  ".eu"

Authors’ reply: Corrected according to suggestions.

Line 298: the link is not correct, you should write again with “Publications” and “HAI-ICU”

Authors’ reply: Corrected according to suggestions.

Line 317: the link is not correct, you should write again with “NHSN/PDFS/DATASTAT/NNIS_2004.PDF

Authors’ reply: Corrected according to suggestions.

Reviewer 3 Report

 Thank you for this interesting article. I only have some minor comments.

Abstract: (Introduction) Abbreviation “HAI” should be defined here because it is used for the first time.

Page 2, line 80: Define BSI

Results: I strongly recommend reducing the number of used abbreviations because they significantly reduce the readability of the manuscript. For example the abbreviations used for various surgical procedures are unnecessary, and can be replaced by the actual terms.

Discussion page 4 lines 164-170: Health may be one of the factors but using less strict criteria for surgical interventions is more likely – please rephrase.

Page 4 line 186. The phrasing is so odd that I found it unclear to what duration this is referring to. Please clarify so that the reader can at once understand that you are talking about the duration of surgical operations here.

Page 5 lines 218-219: Acinetobacter is not a common cause for pneumonia in most countries. If this truly was the case, this should be discussed.  

A clear conclusion is missing. For example, would the authors recommend some methods for other units in order to reduce HAIs?

Author Response

DETAILED RESPONSE TO REVIEWERS,

STEP-BY-STEP REPLIES TO REVIEWERS' COMMENTS:

Reviewer #3,:

Thank you for this interesting article. I only have some minor comments.

Abstract: (Introduction) Abbreviation “HAI” should be defined here because it is used for the first time.

Authors’ reply: Corrected according to suggestions.

Page 2, line 80: Define BSI

Authors’ reply: Corrected according to suggestions.

Results: I strongly recommend reducing the number of used abbreviations because they significantly reduce the readability of the manuscript. For example the abbreviations used for various surgical procedures are unnecessary, and can be replaced by the actual terms.

Authors’ reply: Corrected according to suggestions, significantly reduced the used of abbreviations

Discussion page 4 lines 164-170: Health may be one of the factors but using less strict criteria for surgical interventions is more likely – please rephrase.

Authors’ reply: Corrected according to suggestions, (lines 202-203), as below:

(…) This is probably due to using less strict criteria for surgical interventions or to the health condition of Polish patients who require surgical intervention earlier than is the case in other European countries [16], (…)

Page 4 line 186. The phrasing is so odd that I found it unclear to what duration this is referring to. Please clarify so that the reader can at once understand that you are talking about the duration of surgical operations here.

Authors’ reply: Corrected according to suggestions, (lines 221-222), as below:

(…) This was particularly expressed in the significant reduction in the durations of operations, e.g., in laminectomy; (…)

Page 5 lines 218-219: Acinetobacter is not a common cause for pneumonia in most countries. If this truly was the case, this should be discussed.

Authors’ reply: Corrected according to suggestions, (lines 262-269), as below:

(…) Acinetobacter baumannii is not a common cause of pneumonia in most European countries [32], but in recent years in Polish ICUs, a large burden of pneumonia by Gram-negative bacilli has been documented with dominance of A. baumannii, which was responsible for as many as 1 in 5 cases [31]. A. baumannii is not commonly regarded as a major pneumonia pathogen, and because only a minority of pneumonia microbiological diagnoses were properly made in the studied unit and also in many Polish ICUs, it is difficult to know whether the high prevalence of A. baumannii may have resulted from inadequate diagnosis or environmental contamination, e.g., stethoscopes can be a cause of cross-infection [33]. (…)

A clear conclusion is missing. For example, would the authors recommend some methods for other units in order to reduce HAIs?

Authors’ reply: Corrected according to suggestions, (lines 295-299), as below:

(…) To conclude, the study has validated the usefulness of active surveillance of HAIs in neurosurgical patients based on regular analysis and feedback. We suggest the implementation of infection control and prevention based on evidence-based medicine. Such active SSI surveillance should be recommended by the ECDC for the everyday activity of ICTs and should be adopted by other hospitals in Poland.. (…)

Round 2

Reviewer 1 Report

None